# Transient Receptor Potential (TRP) Channels in Tumor Vascularization

**DOI:** 10.3390/ijms232214253

**Published:** 2022-11-17

**Authors:** Angelica Perna, Carmine Sellitto, Klara Komici, Eleonora Hay, Aldo Rocca, Paolo De Blasiis, Angela Lucariello, Francesco Moccia, Germano Guerra

**Affiliations:** 1Department of Medicine and Health Sciences “Vincenzo Tiberio”, University of Molise, 86100 Campobasso, Italy; 2Department of Mental and Physical Health and Preventive Medicine, Section of Human Anatomy, University of Campania “Luigi Vanvitelli”, 80100 Naples, Italy; 3Department of Sport Sciences and Wellness, University of Naples “Parthenope”, 80100 Naples, Italy; 4Laboratory of General Physiology, Department of Biology and Biotechnology “L. Spallanzani”, University of Pavia, 27100 Pavia, Italy

**Keywords:** transient receptor potential, tumor vascularization, endothelial cells, endothelial colony-forming cells, Ca^2+^ signaling

## Abstract

Tumor diseases are unfortunately quick spreading, even though numerous studies are under way to improve early diagnosis and targeted treatments that take into account both the different characteristics associated with the various tumor types and the conditions of individual patients. In recent years, studies have focused on the role of ion channels in tumor development, as these proteins are involved in several cellular processes relevant to neoplastic transformation. Among all ion channels, many studies have focused on the superfamily of Transient Receptor Potential (TRP) channels, which are non-selective cation channels mediating extracellular Ca^2+^ influx. In this review, we examined the role of different endothelial TRP channel isoforms in tumor vessel formation, a process that is essential in tumor growth and metastasis.

## 1. Introduction

Like all tissues, tumors depend on blood vessels for the supply of oxygen (O_2_) and nutrients, as well as for the removal of metabolic catabolites; blood vessels also make it unavoidable for metastatic cancer cells to spread and invade different districts [1].

Physiologically, endothelial cells (ECs) play a crucial role in the development and maintenance of the vascular network, a process that requires the interaction of three distinct mechanisms: vasculogenesis, angiogenesis, and arteriogenesis [2].

Vasculogenesis is the de novo formation of a vascular network during embryonic development, angiogenesis is a regulated process in which the blood vessel network is formed from pre-existing capillaries, and arteriogenesis refers to the enlargement of pre-existing arterial connections into completely developed and functional arteries. Every day, approximately 100 quiescent ECs undergo turnover in response to pro-angiogenic factors such as vascular endothelial growth factor (VEGF), basic fibroblast growth factor (bFGF), and platelet-derived growth factor (PDGF).

Several processes contribute to neovessel formation in growing tumors and distant metastases: sprouting angiogenesis (SA) which often results in non-productive angiogenesis with blood vessels that are abnormal in structure and function [3,4]; intussusception angiogenesis (IA), which is often activated because it is less energy-intensive [5]; non-angiogenic processes, such as vascular co-option, where the tumor surrounds an already existing vascular network [6]; vasculogenesis, which is mediated by the recruitment of multiple subtypes of endothelial progenitor cells (EPCs), including endothelial colony-forming cells (ECFCs) [7].

The recruitment of many Ca^2+^-sensitive decoders in response to pro-angiogenic cues including VEGF, bFGF, stromal derived factor-1 (SDF-1), and angiopoietins has long been known to take place during angiogenesis. In addition, it has recently been demonstrated that intracellular Ca^2+^ signaling also promotes the proliferation, tube development, and neovessel formation in circulating ECFCs [8,9,10,11]. A crucial role in this process has been attributed to endothelial TRP channels, which are sensitive to pro-angiogenic factors and can regulate both angiogenesis and vasculogenesis [7].

Endogenous Ca^2+^ release through inositol-1,4,5-trisphosphate receptors (InsP3R) and store-operated Ca^2+^ entry (SOCE) through Orai1 channels are the main drivers of the endothelial Ca^2+^ response to pro-angiogenic cues, which can take on a variety of waveforms, including Ca^2+^ transients, biphasic Ca^2+^ signals, and repetitive Ca^2+^ oscillations. Lysosomal Ca^2+^ release via nicotinic acid adenine dinucleotide phosphate (NAADP)-gated two-pore channels is a critical pro-angiogenic route that supports intracellular Ca^2+^ mobilization [12]. Therefore, understanding how TRP-mediated endothelial Ca^2+^ signaling regulates neovessel formation could shed light on alternative strategies to interfere with the aberrant vascularization in cancer.

## 2. Vasculogenesis, Angiogenesis, and Arteriogenesis in Neovessel Formation

Vasculogenesis is the process through which mesoderm-derived progenitors (angioblasts) differentiate into ECs, which then assemble and unite to produce blood islands and the major capillary plexus. Angiogenesis, which can occur by SA or by IA, is primarily responsible for the subsequent extension and remodeling of this primitive network (2). SA is triggered by a tipping of the balance between pro- and anti-angiogenic factors in favor of pro-angiogenic cues. EC activation is induced in response to a gradient of VEGF, which stimulates endothelial tip cells to elongate actin-rich filopodia that migrate toward the source of VEGF, thereby conferring directionality to the sprout [13]. Digestion of the extracellular matrix and basement membrane by specific proteases also occurs during this migration. Behind the tip cells, stalk cells proliferate, allowing the nascent vessel to elongate [13]; this immature vessels is then stabilized by the recruitment of mural cells (i.e., smooth muscle cells and pericytes). IA is a microvascular growth process that tears an existing vessel in two. A cellular pillar is inserted into the arterial lumen, and the whole process seems to depend less on cell migration and more on cell proliferation and reorganization [14]. Finally, arteriogenesis involves pre-existing arterioles that undergo growth and remodeling under the induction of PDGF, and subsequent stabilization through the recruitment of smooth muscle cells (SMCs) or pericytes [13]; the result is the formation of larger vessels [15], as more widely illustrated in the following section.

## 3. Neovessel Formation in the Tumor

Tumor vessels turn out to be highly immature [3], surrounded by only a few pericytes and lacking the adhesion molecule VE-cadherin (vascular endothelial cadherin) with a weakening of the intercellular endothelial junctions. This results in the formation of an endothelium that leads to an increase in interstitial fluid, accumulation of solutes, and transmigration of tumor cells [1,16]. The tumor vasculature is extremely scattered and chaotically distributed, and blood frequently travels via the same channel in multiple directions [17]. Pro-angiogenic factors, such as VEGF, FGF, placental growth factor (PlGF), and angiopoietins, are stimulated even more when there are arteries that are not fully functional, and this results in a continuous cycle of unproductive angiogenesis [4].

In addition to SA, IA is a mechanism that is frequently activated in tumor formation [5] because, from a metabolic standpoint, IA may be less difficult than SA given the low proliferation and migration of ECs along with the quicker rate at which ECM and perivascular cells promote the rapid generation of new vessels. It is known that blocking Notch signaling in the existing vascular bed encourages pericyte detachment and mononuclear cell overflow, which causes rapid vascular expansion via IA, while blocking Notch signaling in the anterior margins of the developing vessel causes SA. The exact molecular mechanisms of IA are still largely unknown [18]. There may be additional non-angiogenic processes for tumor vascularization. One of them is vascular co-option, in which cancer cells take over already-existing, dormant vessels from the surrounding parenchymal tissue and incorporate them into the mass of the tumor. The proliferation of ECs in co-opted vasculature is smaller than that of angiogenic ones, and they exhibit the normal angiogenic markers at low levels. Vessel co-option has been seen in primary and metastatic cancers, particularly in lung, brain, and liver tumors, and is a resistance mechanism to anti-angiogenic therapy. It is associated with a poor prognosis in patients [6].

A crucial contribution to the angiogenic switch that turns a dormant lesion into a growing tumor is provided by the recruitment of myeloid angiogenic cells and ECFCs, which are released from the bone marrow and vascular stem cell niches, respectively. Indeed, several studies have shown that circulating ECFCs contribute to the tumor vasculature in several malignancies, including breast carcinoma [19,20], renal cell carcinoma [19], and melanoma [21], and thus represent a cellular target for dismantling the tumor vasculature [16,22].

## 4. ECFCs in Tumor Vascularization

Tumor vascularization is a far more complicated process than was initially thought. During the development of tumors and the spread of metastatic disease, angiogenic alteration is fueled by the coordinated interaction of local ECs and circulating EPCs [23,24,25]. The cross-talk with the surrounding microenvironment, which is characterized by hypoxia, low pH, disorganized basement membrane, high interstitial fluid pressure, enrichment of growth factors and cytokines, and the different vascular beds of origin, determines the heterogeneity of tumor-endothelial cells (T-ECs) [26].

Tumor-cell-derived VEGF and SDF-1 mobilize EPCs from bone marrow and vascular wall stem cell habitats. Once these mediators are released into circulation, they create a concentration gradient that facilitates EPC recruitment to both the primary tumor lesion and the more remote pre-metastatic niches. Here, EPCs may both physically engraft within tumor vasculature and emit angiogenic factors to drive local angiogenesis, causing the transformation from a latent undetected lesion to a fatal metastatic cancer [25,27,28]. The higher frequency of circulating EPCs corresponds with enhanced angiogenesis and tumor volume and is related with a lower rate of patient survival, which is consistent with the role they play in cancer neovascularization. However, it should be noted that the level of EPC integration within tumor vasculature varies greatly, from 0% to more than 90%. A variety of overlapping factors may be used to account for this heterogeneity [3,23,24]. The kind, stage, and location of the tumor all affect the neovascularization caused by EPC. Mice heterozygous for the tumor suppressor Pten (phosphatase and tensin homolog deleted on chromosome 10) (Pten+/), which can display a variety of malignant neoplasms including uterine carcinomas and lymph hyperplasia, have been used to reveal this trait. In the former, EPC-induced angiogenesis was observed, but not in the latter [29]. EPCs are found in developing pulmonary metastases before vascular formation, implanted LLC (Lewis lung carcinoma), B6RV2 (human lymphoma cells), and melanoma, as well as spontaneous breast cancer occurring in MMTV-PyMT mice (mouse mammary tumor virus-polyoma middle tumor-antigen) [3,23]. They are thereby diminished by both local endothelial cells and hematopoietic cells coming from bone marrow, a characteristic that may understate their contribution at a later stage of tumor growth [24,27]. Last but not least, and maybe the most significant source of variation in the assessment of their role in the angiogenic switch, the EPC phenotype is seldom known [30,31]. The term “EPC” refers to at least two distinct subsets that can be categorized as “hematopoietic” and “non-hematopoietic”, respectively, rather than a single cell population with distinctive surface markers that make it simple to detect and quantify in vivo [31].

The ability of hematopoietic progenitor and stem cells to support vessel growth through the paracrine secretion of growth factors and cytokines, as well as the absence of a combination of markers and receptors selective for truly endothelial EPCs, have both contributed to the confusion surrounding the definition of EPC [28,30]. Colony-forming unit-endothelial cells (CFU-ECs) and circulating angiogenic cells (CACs) make up the hematopoietic EPCs, whereas so-called ECFCs make up the non-hematopoietic EPCs. Despite being drawn to the expanding tumor, CFU-ECs and CACs do not immediately integrate into neovessels because they are located perivascularly and promote malignant growth in a paracrine way. In contrast, ECFCs are devoted to differentiating into mature endothelial cells, form capillary-like structures in vitro, and patent vasculature in vivo [28,30,31]. They also have a remarkable capacity for cloning. They are now recognized as genuine EPCs that have the potential to physically engraft within cancer vasculature. Recent research has proven that human ECFCs, which really come from the endothelium lineage, may be directly injected into the bloodstream of multiple mouse models of human malignancies and then recruited to areas of tumor angiogenesis. More specifically, subcutaneously implanted glioma [32] and breast cancer [33] xenografts, as well as LLC metastases [34], actively reside in DiI-labeled ECFCs. It turns out that ECFC is the best subset to use when examining the molecular pathways causing EPC integration into tumor vasculature and, as a result, when figuring out which targets are most useful in slowing down unfavorable metastatic growth.

## 5. Pro-Angiogenic Ca^2+^ Signals in Vascular Endothelial Cells and ECFCs

Three Ca^2+^ transport systems work together to set the [Ca^2+^]_i_ in vascular endothelial cells between 100 and 200 nM. These systems either extrude Ca^2+^ across the plasma membrane, such as the plasma membrane Ca^2+^-ATPase and the Na^+^/Ca^2+^ exchanger (NCX), or sequester cytosolic Ca^2+^ in the endoplasmic reticulum (ER), the largest intracellular Ca^2+^ reservoir (Sarco-Endoplasmic Reticulum Ca^2+^-ATPase (SERCA)) [35,36]. Neovessel formation depends on an increase in intracellular Ca^2+^ concentration ([Ca^2+^]_i_) in both vascular endothelial cells [11,12,37] and circulating ECFCs. In accord, pro-angiogenic Ca^2+^ signals can be elicited in vascular ECs by a plethora of growth factors, such as VEGF [12], epidermal growth factor [38], and basic fibroblast growth factor, as well as by vasoactive and inflammatory mediators, such as thrombin [39], ATP [40,41], ADP [42], acetylcholine [43], and pleiotropic hormones, including erythropoietin [44]. Pro-angiogenic Ca^2+^ signals can also be delivered to vascular endothelial cells by mechanical stimuli, such as laminar shear stress, which increases during collateral blood vessel growth and stimulates arteriogenesis [38], and endothelial denudation, which stimulates wound repair and endothelial regrowth [45,46]. Furthermore, an increase in [Ca^2+^]_i_ also stimulates proliferation, migration, tube formation and neovessel formation in circulating ECFCs [15,37,47], which represent the only EPC subtype truly belonging to the endothelial lineage [48,49]. For instance, pro-angiogenic Ca^2+^ signals can be induced in circulating ECFCs by VEGF [50], SDF-1α [51], and the human amniotic fluid stem cell secretome [52].

An increase in endothelial [Ca^2+^]_i_ can recruit a number of downstream Ca^2+^-dependent pro-angiogenic decoders, such as the transcription factors nuclear factor of activated T-cells (NFAT), nuclear factor-kappaB (NF-kB), cAMP responsive element binding protein (CREB), myosin light chain kinase (MLCK), myosin 2, endothelial nitric oxide synthase (eNOS), extracellular signal-regulated kinases 1/2 (ERK 1/2), and serine/threonine kinase (Akt) [12].

The endothelial Ca^2+^ response to pro-angiogenic signals can occur as recurrent Ca^2+^ spikes or a transitory or biphasic increase in [Ca^2+^]_i_. Phospholipase Cγ (PLCγ), which cleaves phosphatidylinositol-4,5-bisphosphate (PIP_2_) to generate the second messengers inositol-1,4,5-trisphosphate (InsP_3_) and diacylglycerol (DAG), is brought about by the binding of VEGF to its specific receptor, VEGFR-2, on the plasma membrane. The latter triggers the proto-oncogene, serine/threonine kinase (RAF1)-MEK-ERK1/2 cascade by activating the DAG-sensitive protein-kinase C (PKC). InsP_3_ then activates the InsP_3_R on the ER membrane, thereby causing Ca^2+^ to be released from the ER. The following decrease in ER Ca^2+^ levels induces stromal interaction molecule 1 (Stim1), an ER Ca^2+^ sensor, to oligomerize and translocate from perinuclear to peripheral ER cisternae, where it interacts with the Ca^2+^-permeable channel, Orai1, at ER-plasma membrane junctions known as puncta (10–25 nm). The influx of Ca^2+^ through Orai1 channels has been termed store-operated Ca^2+^ entry (SOCE) and maintains the Ca^2+^ response to VEGF in both vascular endothelial cells [8,12,53] and circulating ECFCs [54]. Concurrent changes in the endothelial membrane potential (Vm), brought about by the recruitment of Ca^2+^-dependent conductances, have the capacity to modify the pro-angiogenic Ca^2+^ signal [55]. Ca^2+^ entry is regulated by the Ca^2+^ driving force, i.e., the difference between the Vm and the equilibrium potential for Ca^2+^ (E_Ca_). The influence of the membrane potential is determined by the expression pattern of various ion channels. K^+^ channels, including Ca^2+^-activated K^+^ channels, inwardly rectifying K^+^ channels, and probably also voltage-dependent K^+^ channels, are the main class of ion channels in the regulation of membrane potential. Furthermore, Ca^2+^ entry into EC requires the presence of extracellular Cl^−^ to maintain a polarized membrane [56]. An important role is attributed to the TRPV4 channel, which is responsible for almost half of the ability to increase [Ca^2+^]_i_ in response to Vm ≥ −60 mV. The remaining [Ca^2+^]_i_ response depends on TRPC1,3,4,5,6/TRPV1,3/TRPA1 [57], which assemble homomeric or heteromeric [58,59].

Several studies have shown that endothelial Ca^2+^ signals may be potentially responsible for aberrant angiogenesis and tumor proliferation [60,61]. The modification of the Ca^2+^ machinery in malignant cells, which contributes to the distinctive hallmarks of cancer [62], is a well-established tenet of neoplastic transformation [63,64]. T-ECs and tumor-derived ECFCs (T-ECFCs) exhibit a significant dysregulation of their Ca^2+^ signaling toolkit [19,65].

## 6. TRP Channels

TRP channels are a large family of cation channels, with sequence homology with the Drosophila TRP channel protein. They are divided into non-selective channels and highly selective cation channels. In mammals, using sequence homology, the 28 members of the TRP channel superfamily of non-selective cation channels can be classified into six subfamilies: TRP Canonical (TRPC1–7), TRP Vanilloid (TRPV1-6), TRP Melastatin (TRPM1–8), TRP Ankyrin1 (TRPA1), TRP Mucolipin (TRPM1-3), and TRP Polycystin (TRPP) [66,67,68,69]. There are eight members of the TRPP subfamily, although only TRPP2, TRPP3, and TRPP5 exhibit the structure and functionality of an ion channel [67]. In addition, TRPC2, which is essential for the mouse acrosomal reaction and pheromone detection, only exists as a pseudo-gene in people [67]. TRP channels can assemble into homomeric or heteromeric channels, combining with subunits belonging to the same or different subfamilies [70,71]. Heteromeric TRP channels, which differ from their homomeric counterparts in terms of biophysical fingerprints and regulation processes, have been extensively documented in naive cells, such as vascular endothelial cells, and heterologous expression systems. The TRPC subfamily has been the subject of in-depth research on subunit heteromerization. For instance, TRPC1 and TRPC3, TRPC4, or TRPC5 [69] may come together to form functional heteromeric channels, but TRPC3, TRPC6, and TRPC7 may also come together to form functional heteromeric channels in both naive tissues [72] and heterologous expression systems [73]. Heterotetramers made up of TRPV subunits are another possibility. Multiple studies have shown that TRPV5/TRPV6 subunits assemble into heteromeric channel complexes [74], as TRPV1-4 subunits. These heteromeric channel complexes are only found on the plasma membrane [58]. Additionally, functional heterotetramers made up of TRP channel subunits from several subfamilies have been widely documented. Examples of these heterotetramers include TRPC1/TRPP2 [70], TRPC1/TRPV4 [59], TRPC1/TRPV6 [75], TRPV4/TRPC6 [76], and TRPML3/TRPV5 [77].

TRP channels are multimodal cellular sensors that can be triggered by a wide range of chemical and physical stimuli, such as intracellular second messengers including diacylglycerol (DAG), arachidonic acid (AA), adenosine diphosphate ribose (ADPr), and hydrogen peroxide (H_2_O_2_), and intracellular ions, such as an increase in cytosolic HC and Ca^2+^, and a decrease in cytosolic Mg^2+^. Other possible triggers include dietary agonists, such as capsaicin, menthol, and allyl isothiocyanate (AITC), synthetic ligands, such as 4α-phorbol-12,13-didecanoate (4αPDD), gasotransmitters, such as nitric oxide (NO), proteins, such as G proteins, mechanical perturbation, such as membrane stretch, osmotic swelling, and laminar shear stress, and temperature fluctuations [67,68,78,79].

Although the relative permeability to Ca^2+^ and Na^+^ (P_Ca_/P_Na_) might vary significantly between the various subunits, TRP channels are permeable to monovalent (Na^+^ and K^+^) and divalent (Ca^2+^ and Mg^2+^) cations. TRP channels transmit inward Na^+^ (and Ca^2+^ or Mg^2+^) currents in response to external stimulation, which have a significant effect on intracellular Ca^2+^ dynamics. In light of this, Ca^2+^ entry via Ca^2+^-permeable TRP channels may directly increase [Ca^2+^]_i_, whereas Na^+^ influx causes membrane depolarization, activating voltage-gated Ca^2+^ channels in excitable cells and altering the driving-force for Ca^2+^ entry in non-excitable cells [67]. Therefore, the “fractional Ca^2+^ current” of Ca^2+^-permeable TRP channels, which has not been measured for all TRP channels, determines the functional effect they have on intracellular Ca^2+^ homeostasis. For instance, TRPA1 and TRPM3 display fractional Ca^2+^ currents of 20%, whereas TRPV1 exhibits a fractional Ca^2+^ current of 5%, in contrast to TRPV5 and TRPV6, which mediate actual Ca^2+^ currents [67,68]. Some TRP channels can cause a significant rise in local Ca^2+^ concentration, which may be spatially restricted to the submembrane space or spread to the bulk cytosol through the Ca^2+^-dependent recruitment of InsP_3_Rs and ryanodine receptors (RyRs) [66,68].

### 6.1. The Role of TRP Channels in Endothelial Cells

ECs line the inner lumen of blood arteries and are consequently exposed to a variety of chemical and physical stimuli (growth factors and chemokines) that must be properly interpreted in order to maintain tissue homeostasis [12,80]. Due to the adaptability of their gating mechanisms, TRP channels provide the most ideal signal transduction system by which vascular ECs integrate such a wide variety of external inputs. Due to their sensitivity to many signaling pathways, endothelial TRP channels, with the exception of TRPC3, TRPC6, and TRPM4, serve as polymodal cellular sensors [66,68].

Vascular ECs have been found to contain the majority of mammalian TRP isoforms (TRPC1-7, TRPV1-4, TRPA1, TRPP1-2 and TRPM1-8) with the exception of TRPM5 [66,68,69], however TRP channel distribution patterns may change across the vascular tree and in various animal species. TRPC1 is present in mouse but not rat aortic ECs [81], whereas TRPC3 is broadly expressed in human but not bovine pulmonary artery ECs. Mouse brain microvascular ECs specifically express TRPC1-6 channels, but not human ones [82,83]; this may be affected by cell culture conditions and expression detection techniques. Naive TRP channels may include heteromeric subunits in vascular endothelial cells, as has been observed in other cell types. The following endothelial TRP channel complexes were reported: TRPC1-TRPC4 [84], TRPC1-TRPV4 [59], TRPC3-TRPC4 [85], TRPV1-TRPV4 [86], TRPP2-TRPC1 [70], and TRPC1-TRPP2-TRPV4 [87].

The majority of ECs functions are regulated by TRP channels, which either create a spatially constrained Ca^2+^ domain around the cytosolic mouth of the channel pore or an increase in [Ca^2+^]_i_ globally [66]; they mediate Ca^2+^ entry in vascular ECs subjected to a variety of acetates, hormones, and mechanical stimuli, such as pulsatile stretch and laminar shear stress [66,68,69]. Additionally, TRP channels influence endothelial V_M_ by conducting a depolarizing inward current carried by Na^+^ and Ca^2+^, resulting in a positive shift in V_M_ that may be amplified by the Ca^2+^-dependent recruitment of TRPM4 (yet to be shown] [66,68,69]. Endothelial TRP channels may either support long-lasting processes, such as gene expression, proliferation, and migration, or short-term responses, such as vasodilation or an increase in vascular permeability. They could also serve as sensors of oxidative stress and local temperature changes; H_2_O_2_ induces an aberrant increase in [Ca^2+^]_i_ by activating TRPM2 in pulmonary artery ECs after ischemic and reperfusion injury in mouse cerebral microcirculation [88]. In mouse aortic ECs, TRPV4 can detect a small increase in temperature (from 19 to 38 °C] but TRPV1 causes Ca^2+^ entry when heating from room temperature to over 40 °C [89]. This sensitivity has been linked to changes in vascular tone and endothelial permeability caused by temperature-dependent changes in NO release [89,90].

### 6.2. The Role of Endothelial TRP Channels in Physiological Angiogenesis

Endothelial TRP channels may facilitate vascular growth by promoting proliferation, migration, and tube formation in response to external growth factors, such as VEGF and bFGF, that are released in peripheral circulation after an ischemic shock or vascular injury [7,12,91]. Alternately, they can stimulate angiogenesis or arteriogenesis, respectively, as a consequence of cellular stress, such as a rise in intracellular ROS or a fall in cytosolic Mg^2+^ levels, or of an increase in laminar shear stress [7,91]. It is also known that angiogenesis is initiated in the hypoxic environment; oxygen concentration is a key regulator of angiogenesis, along with the heterodimeric hypoxia-inducible factor (HIF] protein, which includes HIF1α, degraded under normal conditions, and HIF1β [92].

#### 6.2.1. Role of TRPCs

When DAG is produced in human microvascular endothelial cells (HUVECs) downstream of VEGFR, TRPC3 is activated. This causes Na^+^ to inflow by activating the Na^+^/Ca^2+^ exchanger in a reversal mode, which aids in angiogenesis [93]. In contrast, when there is inhibition of TRPC3 or its silencing with siRNA, VEGF activation of ERK1/2 phosphorylation and stimulation of [Ca^2+^]_i_ transients is attenuated; endothelial tube formation is also suppressed [93]. In EPCs, molecular and pharmacological inhibition of TRPC3 abrogated the Ca^2+^ response induced by VEGF and thus blocked proliferation [94]. Silencing the expression of TRPC3, TRPC4 or TRPC5 also blocked spontaneous [Ca^2+^]_i_ oscillations and inhibited tube formation in HUVEC-derived EA.hy926 cells and in HUVECs [95]. Specifically, TRPC4 silencing attenuated oxLDL-induced proliferation and migration of human coronary ECs and angiogenesis tube formation in vitro on matrigel [96]. TRPC6 also appears to be crucial in human microvascular ECs and HUVECs. Studies using a dominant-negative mutant of TRPC6, with three pore-region alterations, showed decreased EC migration, proliferation, and sprouting in the matrigel experiment [97]. Similarly, in HUVECs, a dominant-negative version of TRPC6 reduced VEGF-evoked capillary formation and cation current, as well as cell growth and proliferation [98]. In EPCs, TRPC1 regulates cell proliferation and tubulogenesis; indeed, an in vivo matrigel experiment showed that EPCs derived from TRPC1 mutant mice had significantly reduced functional activity, including migration and tube formation [99]. The Figure 1A shows the main factors involved in the activation of TRPCs (Figure 1A). Table 1 shows the angiogenic processes regulated by the different subunits [7].

#### 6.2.2. Role of TRPVs

According to a recent study, TRPV4 has long been recognized to control angiogenesis and neovascularization by promoting EC migration and proliferation [60]. TRPV4 is crucial for cytoskeletal remodeling and alterations in cell adhesion, which regulate EC motility and proliferation through mechanotransduction [100,101]. In mutant mice, the lack of TRPV4 was linked to an increase in baseline Rho/Rho kinase activity, a significant increase in EC migration and proliferation, and aberrant tube formation in vitro [101]. Intriguingly, a subsequent investigation from the same team verified that over-expressing or pharmacologically activating TRPV4 with GSK1016790 corrected the aberrant ECs’ abnormal tube formation in the matrigel experiment and restored their mechanosensitivity [100]. TRPV1 has been discovered to promote angiogenesis. Intraperitoneal injection of the TRPV1 ligand evodiamine, promoted vascularization in matrigel plugs used in vivo in wild type mice [102]; in contrast, TRPV1 knockout animals showed a significant reduction in induced angiogenesis. Furthermore, in human microvascular ECs [103], TRPV1 activation is dependent on simvastatin-activated Ca^2+^ influx, which induces activation of CaMKII signaling and enhances TRPV1-eNOS complex formation, leading to NO production and angiogenic tube formation in vitro [104].

The Figure 1B shows the main factors involved in the activation of TRPVs (Figure 1B). Table 2 shows the angiogenic processes regulated of the different subunits [7].

#### 6.2.3. Role of TRPMs

It has also been discovered that TRPM2, TRPM4, and TRPM7 are involved in angiogenesis [105]. VEGF has been shown to stimulate EC migration and induce ROS-dependent Ca^2+^ entry through TRPM2 activation. Furthermore, matrigel plugs supplemented with VEGF injected subcutaneously into TRPM2 knockout mice show significantly reduced blood vessel formation compared to wild type mice [106]. In response to hypoxia/ischemia, TRPM4 is increased in vascular endothelium both in vitro and in vivo, as well as in HUVECs after oxygen-glucose deprivation. Enhancing tube formation on matrigel and improving capillary integrity in vivo were the results of pharmacologically inhibiting TRPM4 or silencing it using siRNA [107]. An earlier study showed that silencing TRPM7 replicates the effect of Mg^2+^ deficit on the development and migration of microvascular ECs, suggesting that magnesium and TRPM7 are regulators of angiogenesis [108]. Figure 1C shows the main factors involved in the activation of TRPMs and TRPA1 (Figure 1C). Table 3 shows the angiogenic processes regulated of the different subunits [7].

A preliminary report showed that only TRPC1 and TRPC4 were expressed by human ECFCs generated from peripheral blood (PB-ECFCs), although TRPC3 was also present in ECFCs produced from umbilical cord blood [109]. Following genetic silencing using specific siRNAs, it was discovered that TRPC1 interacts with STIM1 and Orai1 to produce SOCE in ECFCs [110,111]. It is unclear if Orai1 and TRPC1, which are both gated by STIM1, produce independent Ca^2+^-permeable channels or whether they come together to form a supermolecular complex that may also include TRPC4. SOCE can be recruited by SDF-1α, to increase ECFC migration in vitro and neovessel development in vivo [10], as well as by VEGF to promote ECFC proliferation and tube formation [54,112]. Specifically, whereas the pro-angiogenic response to VEGF requires the Ca^2+^-sensitive transcription factor NF-kB, SOCE activates the ERK1/2 and PI3K/Akt signaling pathways to promote SDF-1α-induced ECFC motility [10,52]. STIM1, TRPC1, and Orai1 interact to drive SOCE and facilitate in vitro angiogenesis (proliferation, motility, and formation) in rodent MACs [113], just as they do in human ECFCs. Furthermore, CaM-dependent e NOS activation was impaired by genetic TRPC1 knockdown, which hindered neovessel formation in Matrigel plugs in vivo [99]. TRPC3 is only produced in umbilical cord blood (UCB)-derived ECFCs, where it is physiologically gated by DAG and causes intracellular Ca^2+^ oscillations generated by VEGF [94]. Genetic (using certain siRNAs) and pharmacological (using Pyr3) treatments demonstrated that TRPC3-mediated extracellular Ca^2+^ entry causes the dynamic interaction between InsP_3_Rs and SOCE to alter the spiking Ca^2+^ response to VEGF, increasing the proliferation of UCB-ECFC [94]. The increased frequency of VEGF-induced intracellular Ca^2+^ oscillations in UCB-ECFCs, which is related to their higher proliferative capability, has been postulated to be caused by TRPC3 participation [114]. This finding gave rise to the theory that exogenous TRPC3 insertion could rejuvenate the reparative phenotype of senescent/aging UCB-ECFCs and increase the effectiveness of autologous cell-based therapy in ischemic patients [114].

The two most significant endothelial TRPV isoforms involved in angiogenesis and arteriogenesis are TRPV1 and TRPV4. Multiple independent studies revealed that human ECFCs also express TRPM7 [108], TRPV1 [115], and TRPV4 [116]. Pharmacological stimulation of TRPV4 with the endogenous agonist, arachidonic acid, stimulated PB-ECFC proliferation in a NO-dependent manner [117], while genetic silencing of TRPM7 with a specific siRNA had no impact on the rate of ECFC proliferation [108]. The activation of TRPV1 could also be sufficient to cause ECFC proliferation [91,118]. Previous work has shown that TRPV1 promoted proliferation of UCB-ECFC cells, as well as of the HUVEC-derived EA.hy926 cells, by mediating anandamide uptake independently of Ca^2+^ entry [119]. Subsequently, Lodola et al. (2019) [120] found that optical excitation of PB-ECFCs plated on the photosensitive polymer Poly(3-hexyl-thiophene) (P3HT) stimulated proliferation and tube formation by stimulating the nuclear translocation of p65 NF-kB [120]. A recent investigation showed that TRPV1 activation depends on the local increase in ROS levels at the interface between P3HT thin layers and the cell surface [115].

## 7. The Role of Endothelial TRP Channels in Tumor Vascularization

In a growing number of cancers, recent data have suggested that deregulation of the endothelial Ca^2+^ mechanism is essential for neovascularization and resistance to anti-angiogenic and chemotherapeutic treatments [41,60,61]. A relevant role has been attributed to abnormal expression and activity of TRP channels in T-ECs and T-ECFCs, which can boost tumor neovascularization.

### 7.1. Breast Cancer

TRPV4 has been the first endothelial TRP channel directly linked to malignant angiogenesis. In line with this, TRPV4 expression was noticeably increased in breast-tumor-derived endothelial cells (B-TECs), and TRPV4 activation with AA or 4αPDD stimulated migration in B-TECs, but not in the control human microvascular endothelial cells. On the other hand, when cells were transfected with a short hairpin RNA that selectively targets TRPV4 (shTRPV4), AA-induced B-TEC migration was stopped [121]. Notably, a brief (10 min) pre-incubation with AA alone dramatically raised the cell surface expression of TRPV4, thereby increasing AA-induced extracellular Ca^2+^ entry in migrating B-TECs compared to non-migrating cells. Furthermore, AA has been shown to stimulate extracellular Ca^2+^ entry in B-TECs during the first stages of the tubulogenic process, but not when a capillary-like network was already developed. These findings clearly suggest that the early stages of breast cancer angiogenesis are characterized by the over-expression of endothelial TRPV4 channels [122] (Figure 2).

TRPM8 was also found to control breast cancer angiogenesis, but in a Ca^2+^-independent manner. In contrast to HMECs and HUVECs, in which it was mostly found in the ER, TRPM8 was substantially down-regulated in B-TECs [123]. Intriguingly, in normal endothelial cells, TRPM8 prevented migration and tube formation by trapping Rap1 intracellularly and blocking its movement towards the plasma membrane, which is essential to trigger β1-integrin signaling. As a consequence, TRPM8 down-regulation in B-TECs is likely to accelerate vascular growth; this feature suggests that TRPM8 activation by icilin or menthol could represent an efficient strategy to treat breast cancer [123] (Figure 2).

TRPM8 is down-regulated in B-TEC, Rap1 isn’t trapped, β1-integrin isn’t inhibited, migration and tube formation are blocked.

### 7.2. Prostate Cancer

Prostate cancer (PCa) involves a complex remodeling of endothelial TRP channels. TRPA1, TRPV2, and TRPC3 were shown to be up-regulated in three different endothelial cell lines established from PCa patients as well as in endothelial cells lining tumor capillaries in vivo [124]. In PCa-derived endothelial cells, it has been reported that: (1) TRPA1 supports migration; (2) TRPC3 supports chemoattraction towards tumor microenvironment; and (3) TRPV2 induces capillary-like formation in vitro. Furthermore, TRPV2 activation stimulates vascular development in a mouse model of postnatal retina in vivo [124]. When compared to non-tumor endothelium, the pro-angiogenic effect of TRP channels on PCa-derived endothelial cells was accompanied by a significant increase in intracellular Ca^2+^ activity. As a result, PCa patients could benefit from the pharmacological blockage of these specific endothelial TRP channel isoforms [124].

A parallel investigation showed that TRPV4 was down-regulated in prostate adenocarcinoma-derived endothelial cells (A-TECs), thereby reducing their mechanosensitivity to extracellular matrix (ECM) rigidity. This, in turn, favored A-TEC motility. The tumor vasculature in TRPV4 KO mice was characterized by an increased proportion of hyperpermeable, pericyte-free and dilated microvessels, which are known to moderate the therapeutic effect of anticancer treatments [100]. A subsequent report found that TRPV4 down-regulation significantly reduced the VE-cadherin expression at cell–cell contacts, thereby further increasing vascular leakage [125]. To rescue the phenotype of aberrant capillary tubules in vitro, overexpression or pharmacological stimulation of TRPV4 with GSK was sufficient to restore their mechanosensitivity to ECM rigidity by blocking basal Rho activity [126]. Furthermore, TRPV4-mediated extracellular Ca^2+^ influx blocked the ERK1/2 pathway, reducing the rate of A-TEC proliferation (Figure 3). These results imply that TRPV4 inhibits the development of malignant vasculature in the adenocarcinoma of the prostate [100,126].

### 7.3. Renal Cellular Carcinoma

The most common form of kidney cancer in adults is renal cellular carcinoma (RCC) and there is substantial evidence that ECFCs may play a primary role in RCC neovascularization [25,127,128,129,130,131]. All research carried out on the subject has supported each other in demonstrating that ECFCs need functional VEGFR-2 to maintain malignant transformation, although they reached this conclusion by using normal cells rather than tumor cells [16,65]. RCC has recently been linked to a significant deregulation of the Ca^2+^ signaling machinery; in accord, primary tumor samples express more Orai1 and TRPC6 channel proteins than normal renal tissues [132,133]. In contrast, two separate human kidney cancer cell lines exhibited TRPC4 down-regulation [134]. These changes might be crucial to the neoplastic transformation of a healthy kidney. While Orai1 controls RCC cell proliferation and migration [133] and TRPC6 up-regulation favors the transition through the G2/M phase [132], the loss of TRPC4 results in a decreased release of the endogenous inhibitor thrombospondin-1, which favors the angiogenic switch [134]. In addition, ECFCs derived from naive RCC patients (RCC-ECFCs) exhibit a striking decrease in ER Ca^2+^ concentration and InsP_3_R expression, while they displayed a remarkable up-regulation of Stim1, Orai1, and TRPC1. These changes result in a remarkable rewiring of their Ca^2+^ toolkit, which render RCC-ECFCs less responsive to VEGF [111]. However, the pharmacological blockade of SOCE has been shown to interfere with RCC-ECFC proliferation and tube formation [25] (Figure 4).

## 8. Conclusions

Endothelial TRP channels play a crucial role in vascular remodeling, regulating angiogenesis, arteriogenesis and vasculogenesis. These channels are highly heterogeneous in their gating mechanisms, with the propensity of some isoforms to form heteromeric complexes, making them the most versatile Ca^2+^ entry pathway in ECs, EPCs, and ECFCs. However, they can also block angiogenesis, as reported for TRPM4 in HUVECs and TRPM7 in cerebral microvascular endothelial cells. An important aspect of endothelial TRP signaling that deserves much attention is its involvement in the abnormal vascularization that characterizes cancer. A crucial contribution to the angiogenic switch that transforms a dormant lesion into a growing tumor is provided by the recruitment of myeloid angiogenic cells and ECFCs, which are released from vascular stem cell niches, respectively. The abnormal expression and activity of TRP channels in T-ECs and T-ECFCs, may promote tumor neovascularization, however, several aspects are still not entirely clear, e.g., which endothelial TRP isoforms are dysregulated, and therefore further studies are needed.

## Figures and Tables

**Figure 1 ijms-23-14253-f001:**
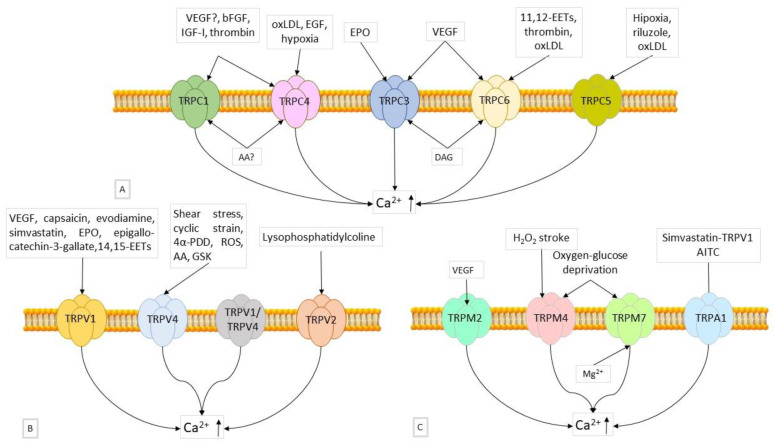
(**A**) TRPC channels in angiogenesis. Representation of the main TRPC channels involved in angiogenesis. The factors that stimulate the expression of the different components are shown. VEGF, IGF, bFGF, and thrombin induce expression of TRPC1; EGF, oxLDL, and hypoxic condition induce expression of TRPC4; erythropoietin (EPO) and VEGF induce expression of TRPC3; oxLDL, thrombin, and 14,15-EETs induce expression of TRPC6; hypoxic condition, riluzole, and oxLDL induce expression of TRPC5. (**B**) TRPV channels in angiogenesis. TRPV1, TRPV2, and TRPV4 are the main TRPV isoforms that mediate angiogenesis. VEGF, capsicin, evodiamine, simvastatin, EPO, epigallo-catechin-3-gallate, and 14,15-EETs induce TRPV1 activation; Shear stress, cyclic strain, 4α-PDD, ROS, AA, and GSK activate TRPV4; lysophosphatidylcholine can induce TRPV2 stimulation. (**C**) TRPM and TRPA1 channels in angiogenesis. It has been shown that certain TRPM isoforms are involved in angiogenesis. VEGF stimulates TRPM2; H_2_O_2_, pathological conditions such as stroke, oxygen, and glucose deprivation induce TRPM4; conditions of reduced intracellular Mg^2+^ concentration, oxygen, and glucose deprivation induce TRPM7 activation; AITC and simvastatin-dependent TRPV1 activation induces TRPA1.

**Figure 2 ijms-23-14253-f002:**
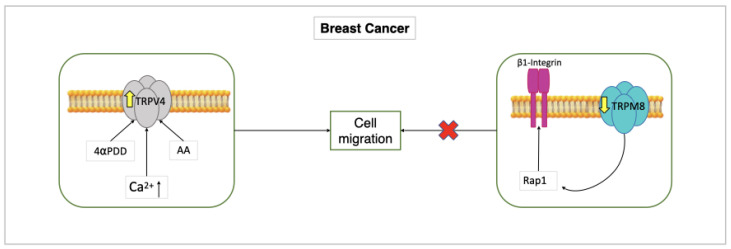
Role of TRPV4 and TRPM8 in B-TECs. AA or 4αPDD induce increased expression of TRPV4 in B-TECs, resulting in increased migration of these and increased intracellular Ca^2+^ entry. This shows that over-expression of TRPV4 induces the early stages of tumor angiogenesis.

**Figure 3 ijms-23-14253-f003:**
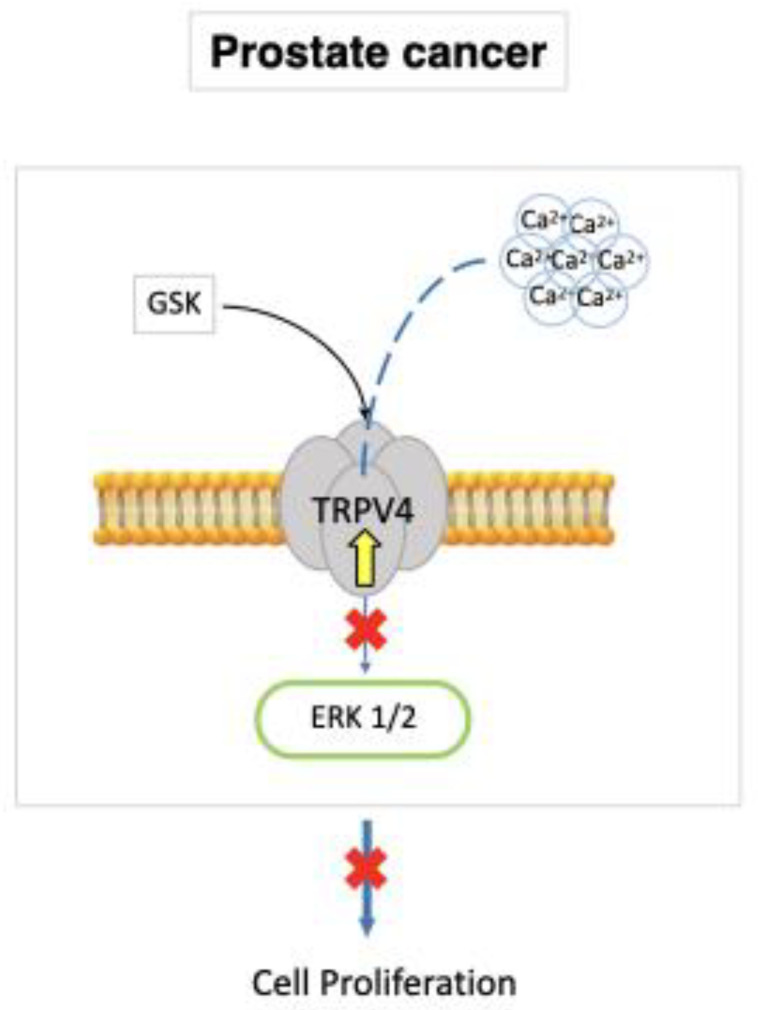
TRPV4 is up-regulated in A-TECs. Pharmacological GSK stimulation of TRPV4 restores mechanosensitivity to ECM rigidity, blocks the ERK1/2 pathway, and reduces the rate of A-TEC proliferation.

**Figure 4 ijms-23-14253-f004:**
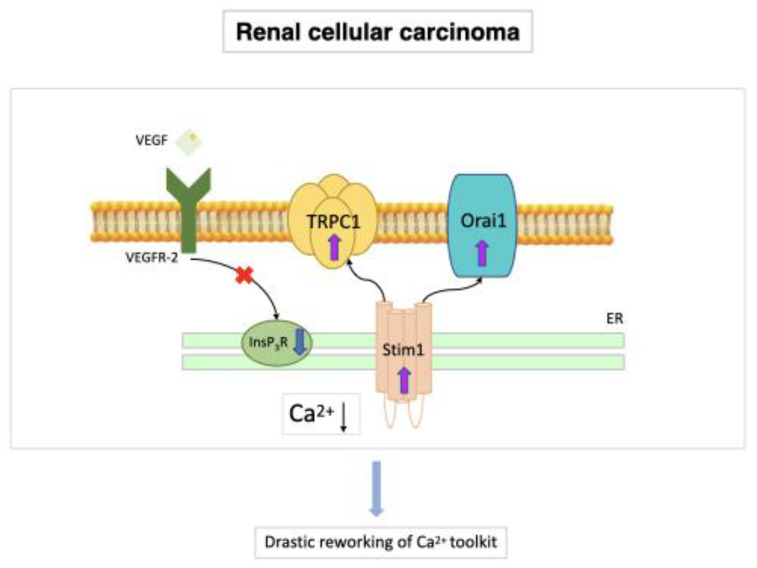
Stim1, Orai1, and TRPC1 (SOCE) in RCC-ECFCs. RCC-ECFCs show a significant decrease in ER Ca^2+^ concentration and InsP_3_R expression, while they present higher expression levels of Stim1, Orai1, and TRPC1 compared to control ECFCs. This results in a dramatic rewiring of their Ca^2+^ signaling toolkit. The higher amplitude of SOCE in RCC-ECFCs could underlie the increase in frequency reported in peripheral blood of RCC patients in response to hypoxic cues released by the tumor microenvironment [19,65].

**Table 1 ijms-23-14253-t001:** Angiogenic processes regulated by TRPC subunits.

TRPC Subunit	Angiogenic Processes Regulated
**TRPC1/TRPC4/TRPC3/TRPC5/TRPC6**	Proliferation, migration, in vitro tubulogenesis.
**TRPC1**	Filipodia extension, motility in vivo (sprouting angiogenesis); ECFC and MAC proliferation and tube formation.
**TRPC4**	Retinal neovascularization.
**TRPC6**	Wound closure in vitro and carotid artery regeneration in vivo.
**TRPC5**	Neovascularization in hypoxic retina and in mouse hindlimb ischemia. Negative modulation of migration and wound closure in vitro and arterial regeneration in vivo.

**Table 2 ijms-23-14253-t002:** Angiogenic processes regulated by TRPV subunits.

TRP Subunit	Angiogenic Processes Regulated
**TRPV1**	Proliferation, migration and tube formation in vitro, angiogenesis in vivo.
**TRPV4**	Proliferation, migration, tube formation in vitro, angiogenesis and arteriogenesis in vivo, ECFC proliferation.

**Table 3 ijms-23-14253-t003:** Angiogenic processes regulated by TRPM subunits and TRPA1.

TRP Subunit	Angiogenic Processes Regulated
**TRPM2**	Migration in vitro, neovascularization in vivo.
**TRPM4**	Negative regulation of in vitro tubulogenesis and in vivo angiogenesis; supports H_2_O_2-_induced migration.
**TRPM7**	Negative regulation of HUVEC proliferation, adhesion, and migration in vitro and tubulogenesis in vivo; positive regulation of HMEC proliferation and migration.
**TRPA1**	Tube formation in vitro and neovascularization upon corneal cauterization in vivo.

## Data Availability

Study did not report any data.

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
