# Peer review of "Transient Receptor Potential (TRP) Channels in Tumor Vascularization"

_ijms, 2022, doi:10.3390/ijms232214253_

Round 1

Reviewer 1 Report

This review article provides a probably very useful survey of an extensive literature, and if properly presented would be of value. However, it is very difficult to follow; I kept losing track of the many abbreviations, and the relation among them. In addition, in the end I did not understand where the Ca2+ went, and which channels moved Ca2+ where. Which process produced vascularization, or angiogenesis? And, most important, which cancers went with which process? Is there a pattern, or do we have just a collection of channels that seem to produce new blood vessels, by whichever process? What is needed to make this a useful paper are several more diagrams, which produce a track following the Ca2+ from its store, through which channel, to produce whichever form of new blood vessels, in which cancer. The same may done for each relevant substance transmitted through the TRP channels. I was not able to keep track of what signals, Ca2+ or otherwise, led to which blood vessel forming process, via which channels. The entire value of this review is in its review of the literature to pull together the findings of many authors into a coherent pattern—but the pattern, if any, is not shown in this presentation. In addition, is there anything known about the structure of ECFCs? Is this relevant, and if not, how is it possible that it is not? Can a diagram, or photomicrograph, make this clear? It seems that it must be relevant, from the text, but it is hard to tell just how.

The authors have done half a useful job, in collecting the information. The second half would consist in pulling it together to show what it means, if there is enough information to do this. If not, the places in the paths that need to be filled in can be specified, and this review might be able to help steer research to where it is needed. As it is, the paper is so difficult to read that it will be largely ignored, and if it is simply left as a compilation of references, it will deserve to be. However, it appears that the information needed to define the ways in which TRP channels are involved in transmitting information as signals for expansion of blood vessel formation is present in this paper, and it would be a very useful paper if the information were used to work out the implicit pathways that are buried in the text without being pulled into coherent form.

Author Response

Dear Reviewer,

Thank you for your revision work on the manuscript.

Taking all comments into account, I have completely reorganized the text of the manuscript trying to improve the understanding of the various concepts. I have added more specific paragraphs explaining separately the process of neovascularization, in physiological state and tumor, the involvement of Ca2+ and TRP channels specifically involved in these processes. I have explained the role of ECFCs and the expression of TRP channels in these cells, bringing this into the tumor vasculature. I modified the pictures and made a diagram summarizing the most important mechanisms.

I think the manuscript will be now clearer so to stimulate and further studies. I hope it can now be considered for publication.

Reviewer 2 Report

Review: Transient Receptor Potential (TRP) channels in tumor vascularization have an interesting and up-to-date topic, however the review itself, its structure needs a significant reorganization and improvement.

My suggestions are as follows:

Line 91: Tumor vascularization: angiogenesis and vasculogenesis – chapter – is generally about embryonic vasculature development, therefore too general and already covered by many reviews – this chapter should be specifically and briefly about tumor angiogenesis.

Fig 1 – also too general covered elsewhere. May be transformed in scheme, which factors are involved in tumor anigogenesis. But this is also covered elsewhere, so I better suggest taking it out of the manuscript.

Line 145 – chapter 3:  TRP channels. Again, too general chapter covered in many reviews, can be mentioned as citations to other reviews. Should cover only TRPs members involved in tumor angiogenesis, not all members of whole superfamily.

Figure 2.- also too general. Not focused on tumor signaling.

Line 197 – chapter 4.: Role of endothelial Ca2+ signaling in vessel formation – interesting topic but needs reorganization. Also, better focus on Ca2+ signaling and its effect on endothelial cell and vessels formation. Needs to include information not only from in vitro cultured EC, but also from embryo/tumor/organ angiogenesis.

Schemes illustrating Ca2+ signaling cascade, and scheme about Ca2+ flow of various TRPs in angiognesis activation, will be beneficial.

Parts addressing individual TRPs and its effect on tumor angiogenesis are interesting and beneficial.

In my opinion manuscript needs significant reorganization, text and figures improvements, therefore I do not recommend the manuscript to be published in this form.

Author Response

Dear Reviewer,

Taking your suggestions into account, I reviewed the manuscript, trying to make it clearer in its comprehension.

- Line 91: Tumor vascularization: angiogenesis and vasculogenesis – chapter – is generally about embryonic vasculature development, therefore too general and already covered by many reviews – this chapter should be specifically and briefly about tumor angiogenesis.

Response

I explained the process of neovascularization with a brief introduction of the physiological processes, then focused on the involvement of Ca2+ and specific TRP channels expressed in ECFCs, involved in tumor vascularization.

- Fig 1 – also too general covered elsewhere. May be transformed in scheme, which factors are involved in tumor anigogenesis. But this is also covered elsewhere, so I better suggest taking it out of the manuscript.

Response

I have removed the image.

- Line 145 – chapter 3:  TRP channels. Again, too general chapter covered in many reviews, can be mentioned as citations to other reviews. Should cover only TRPs members involved in tumor angiogenesis, not all members of whole superfamily.

Response

After a brief introduction I have focused on describing only the endothelial TRP channels involved in the tumor angiogenic process.

- Figure 2.- also too general. Not focused on tumor signaling.

Response

I have deleted the image

- Line 197 – chapter 4.: Role of endothelial Ca2+ signaling in vessel formation – interesting topic but needs reorganization. Also, better focus on Ca2+ signaling and its effect on endothelial cell and vessels formation. Needs to include information not only from in vitro cultured EC, but also from embryo/tumor/organ angiogenesis.

Response

I have reorganized the paragraph dealing with Ca2+ signaling in vessel formation, describing the various processes more specifically, focusing on the ECFCs directly involved.

- Schemes illustrating Ca2+ signaling cascade, and scheme about Ca2+ flow of various TRPs in angiognesis activation, will be beneficial.

Response

I have added new pictures to make the mechanisms more understandable.

I think the manuscript will be now clearer so to stimulate and further studies. I hope it can now be considered for publication.

Reviewer 3 Report

This review includes interesting insights on TRP channels in tumor vascularization; however, the current version is like a draft. All authors must carefully read and amend this manuscript. For incomplete revision, the reviewer will judge rigorously.

Major concerns:

1.       All figures lack impact. The authors MUST change into impressive ones.

2.       I can easily find many mistakes (below and the others including sentences).

For instance,

Line 107: Remove ‘(MMPs)’.

Both IP3 and InsP3 are used.

Line 139: Remove ‘endothelial colony forming cells’.

Line 172: (STIM1)

Line 175: so-called store-operated Ca2+ entry

Line 201: TRP o PIEZO - ???

Line 202: Remove ‘so-called store-operated Ca2+ entry’.

Line 203: Remove ‘endoplasmic reticulum’.

Line 207: Remove ‘(SOC)’.

Line 210: SOCE

Line 213: Remove ‘of’.

Line 215: Remove ‘basic fibroblast growth factor’.

   Line 218 and 232: NF-κB

Line 227: SOC channels

Line 249 and 250: Remove ‘(MAMs)’ and ‘(OMM)’.

…..

Line 434: a mouse prostate adrenocarcinoma (TRAMP) model >>> transgenic adenocarcinoma of mouse prostate (TRAMP) model

Author Response

Dear Reviewer,

Thank you for your revision work on the manuscript.

Taking all comments into account, I have completely reorganized the text of the manuscript trying to improve the understanding of the various concepts.

- 1.  All figures lack impact. The authors MUST change into impressive ones.

Response

All images have been removed and new images have been made, trying to increase their impact.

- 2. I can easily find many mistakes (below and the others including sentences).

Response

The text has been completely reworked and all abbreviation and conceptual errors have been corrected.

I have added more specific paragraphs explaining separately the process of neovascularization, in in physiological state and tumor, the role of Ca2+ and TRP channels specifically involved in these processes. I have explained the role of ECFCs and the expression of TRP channels in these cells, bringing this into the tumor vasculature. I modified the pictures and constructed a diagram summarizing the most important mechanisms.

I think the manuscript will be now clearer so to stimulate and further studies. I hope it can now be considered for publication.

Round 2

Reviewer 1 Report

Overall, this a useful review of an extensive literature, and it accumulates a large amount of work in a small space. There are still points where it was too condensed to make comprehension straightforward. However, it is principally a compilation and systematization (to some extent) of the extensive literature on the subject, and it mostly succeeds.

This manuscript is a considerable improvement over the initial form of the manuscript; at least most of the abbreviations were defined. Unfortunately, a few seem to slipped in without definition. For example, AA appears in the caption to Fig.1, and then several times thereafter, without ever being defined. The same is true for 4αPDD—possibly fewer repetitions of this. These are not the only examples.

The Tables and Figures help considerably in organizing the material. The captions for the figures are given at the end, but the Tables have to be labeled above the Table. The English is mostly acceptable (except for some disagreements of singular and plural, and some cases of pronouns with uncertain antecedents) through section 6; some sentences need to be split into two, or at least broken with a semicolon. The second sentence in section 4 is an example. From the beginning of Section 7 on, major corrections are required in the use of English.

The manuscript is, on the whole, acceptable, with these corrections.

Author Response

Dear Reviewer,

in accordance with your comments, I have rechecked the abbreviations, adding definitions where missing. I have added labels to the tables and rechecked the English of the manuscript. I have adjusted the second sentence in section 4 and made the necessary corrections from section 7.

I hope it can now be considered for publication.

Reviewer 3 Report

The authors addressed all comments and revised the manuscript accordingly. I have no more concerns.

Author Response

Dear Reviewer,

thank you for your revision work on the manuscript and for considering the manuscript suitable for publication.